# The Complex Relationship between Mechanisms Underlying Inflammatory Bowel Disease, Its Treatment, and the Risk of Lymphomas: A Comprehensive Review

**DOI:** 10.3390/ijms25084241

**Published:** 2024-04-11

**Authors:** Katarzyna Stasik, Rafał Filip

**Affiliations:** 1Department of Gastroenterology with IBD Unit, Clinical Hospital No. 2, 35-301 Rzeszow, Poland; kstasik@ur.edu.pl; 2Faculty of Medicine, University of Rzeszow, 35-959 Rzeszow, Poland

**Keywords:** Crohn’s disease, inflammatory bowel disease, lymphoma, Hodgkin lymphoma

## Abstract

Patients with inflammatory bowel disease may be at higher risk of developing lymphomas and other cancers of the gastrointestinal tract. In addition, there may be a link between the use of thiopurines or anti-tumor necrosis factor drugs (anti-TNF) and these pathologies. The treatment of patients with Crohn’s disease who have previously been diagnosed with lymphoma is a challenge for gastroenterologists. In this report, we examine important clinical issues related to the treatment of patients with inflammatory bowel disease with active lymphoma, as well as of patients with hematological cancer history. In this discussion, we take into account most of the available treatments for inflammatory bowel disease, as well as the impact of chronic inflammation and viral infections. In addition, we try to find common ground for the development of lymphoproliferative disorders and autoimmune diseases. Patients with inflammatory bowel disease may be at higher risk of developing lymphomas and other cancers of the gastrointestinal tract. Chronic inflammatory processes and viral infections play an important role in carcinogenesis. In addition, there may be a link between the use of thiopurines or anti-TNF drugs and these pathologies. A significant risk of the development of lymphoma in people undergoing each therapy should be considered, and it should be estimated how much greater this risk will be in patients with a history of lymphoproliferative disorders. The following review is an attempt to answer which therapy would be the most appropriate for patients with Crohn’s disease and a history of lymphoma treatment. A lack of clear guidelines creates great challenges for doctors.

## 1. Introduction

Inflammatory bowel disease (IBD), with two subtypes, i.e., Crohn’s disease (CD) and ulcerative colitis (UC), is characterized by chronic IBD. The etiology of these conditions involves interactions between the environment, the immune system, the gut microbiome, and a genetic predisposition to disease [1,2].

UC is a disease with relapses and remissions, and the incidence in the population continues to rise. It is characterized by mucosal inflammation that begins distally and can extend proximally, involving the entire colon. It manifests with bloody diarrhea, abdominal pain, fatigue, and fecal incontinence [2].

CD as opposed to UC is a transmural granulomatous inflammation. Lesions can involve any part of the gastrointestinal tract from the mouth to the rectum. The course of the disease is often complicated by abscesses, intestinal fistulas, intestinal–cutaneous fistulas, intestinal–vaginal fistulas, or intestinal–bladder fistulas [3,4].

IBD also manifests with other extraintestinal manifestations, i.e., anemia, arthropathy (axial or peripheral), cutaneous (erythema) nodosum, purulent gangrenous dermatitis, or ocular symptoms (anterior uveitis) [1].

Lymphomas are a diverse group of lymphoproliferative diseases (LPD) that differ with respect to the origin of lymphocytes, clinical symptoms, and prognosis. Lymphomas arise from lymphocytes at various stages of development, and the characteristics of a particular lymphoma subtype reflect the characteristics of the cell from which they originate [5]. Several studies have reported an association of certain autoimmune diseases with an increased risk of non-Hodgkin lymphoma (NHL) [6,7]. The association with autoimmune disorders has also been found to be stronger in aggressive NHL [8].

In NHL patients, most (70%) autoimmune diseases preceded the onset of lymphoma; however, in Hodgkin lymphoma (HL), autoimmunity developed mainly after treatment of the malignancy [9]. For this reason, considering both processes and their common mechanisms, one should think about how to treat patients with a history of lymphoma.

In patients with IBD, carcinogenesis factors may include chronic inflammation due to the chronic self-stimulation of the immune system [10,11], as well as secondary immune disorders caused by immunosuppressants [12].

## 2. Cancer in Inflammatory Bowel Disease

When considering the processes that contribute to the development of lymphomas, it should be noted that patients with IBD have a higher incidence of cancers in various locations. First of all, the risk of developing cancers of the gastrointestinal tract increases, and colorectal cancer is definitely more common, which has been reported in the literature for many years [13].

The development mechanism of colorectal cancer differs from sporadic cancer in the general population. Tumorigenesis in patients with IBD is in accordance with the definition of chronic inflammatory processes and leads to dysplasia. The release of proinflammatory mediators during IBD leads to developing malignancy by increased oxidative stress that favors DNA mutagenesis, antiapoptotic pathways, angiogenesis, and invasion [14].

Risk factors here are the duration of the disease as well as the extent to which the inflammatory lesions cover [15]. Another well-studied risk factor is the presence of primary sclerosing cholangitis (PSC) [15,16]. PSC has been shown to increase the risk of both colorectal cancer and biliary tract cancer. The risk of cancer is estimated to increase linearly from year to year, after 6–8 years of disease duration. This has led to the formulation of relevant guidelines for performing colonoscopy in IBD patients by ECCO—the European Crohn’s and Colitis Organisation [16].

There are many reports about the risk of developing cancer in patients with IBD. Many potential risk factors are considered. In the work of Biancone et al., they determined the most common types of cancer in patients with IBD, noting that lymphomas and leukemias were observed only in patients with CD (Figure 1) [17].

IBD patients are at increased risk for gastrointestinal tract neoplasms, in particular, adenocarcinomas and small bowel lymphomas, which are usually difficult to diagnose since their symptoms may overlap with the exacerbation of IBD [18,19].

## 3. Lymphoma in Inflammatory Bowel Disease

Despite the many differences between CD and UC, the mechanisms underlying the development of lymphoma in these patient groups appear likely to be similar. However, although based on observational studies, the risk of developing lymphoma appears to be higher in patients with CD, this may be related to both the inflammatory process, which is more difficult to control, and the more frequent use of immunosuppressive therapy [16,17]. In the absence of immunosuppressive therapy, the risk of developing lymphoma is comparable to that of the general population. In addition to the immunosuppressive therapy used, viral infections, organ transplantation, history of chemotherapy, and history of radiation therapy are also risk factors [20].

An increased risk for developing HL or NHL in the course of IBD is well documented in the current literature, and this risk is greater if the disease is diagnosed before the age of 25 years [21]. This may result from a chronic inflammatory condition [22,23,24] or therapy, mainly immunosuppression and/or biological therapy [21,22,23,24,25,26,27,28,29]. Moreover, hematological disorders such as anemia, leukocytosis and thrombocytosis, hypercoagulability, and acute or chronic forms of leukemia have been reported in IBD patients. Some data indicate that the absolute index of lymphomas (HL, NHL, and chronic lymphocytic leukemia (CLL)) is 1.1% for UC and 0.7% for patients with CD; however, pharmacotherapy was not analyzed in this study [30].

Whether IBD itself predisposes to lymphoma is still unclear. Although the increased risk for adenocarcinomas in both UC and colonic CD is well recognized, the situation regarding lymphoproliferative neoplasms remains unclear. Lewis et al. have acknowledged that the relative risk of developing lymphoma in patients with CD or UC did not significantly increase when compared to the general population [28]. On the other hand, other studies involving the Manitoba IBD cohort showed an increased risk of developing lymphoma in this group of patients [30,31].

The most common type of lymphoma in IBD patients is NHL diffuse B-cell lymphoma, 60% of which are nodal; however, it may also involve extranodal structures. The next most common types include follicular lymphoma and HL [10,32,33]. Primary mediastinal large B-cell lymphoma (PMBCL) is approximately 3–5% of cases, especially in young adults (median age 35). It typically presents as a large tumor of the mediastinum, often with infiltration of the surrounding tissues, and does not damage the lymph nodes [34]. The greater incidence of B-cell lymphomas may be explained by the action mechanism of immunosuppressive and biological drugs. By inhibiting cytotoxic T-lymphocytes, the unregulated proliferation of B-lymphocytes occurs. This has been demonstrated in Epstein–Barr virus (EBV)-infected B-cells [35,36].

It can be assumed that diffuse large B-cell lymphoma (DLBCL) would be the most common type of lymphoma in IBD patients, as they have a similar localization to DLBCL found in the general population. However, if chronic inflammation were the underlying cause, it appears logical that lymphoma would be more common in the gastrointestinal tract in patients with CD (Figure 2) [10,11,32].

However, most studies have not confirmed this increased risk. Because of this, it appears that the risk of developing lymphoma in IBD patients is similar to that of the general population [30]. The table below reviews the literature from 2010 to 2022 regarding data related to the incidence of lymphoma surveillance in patients with IBD (Table 1).

Nevertheless, guidelines for treating severe CD in patients with a history of lymphoproliferative neoplasms have not yet been established, and therefore, treatment options in these patients are very limited.

## 4. Characteristic Risk Factors in Patients with CD—Mechanisms Underlying the Development of Lymphomas

### 4.1. Role of the Inflammatory Process in the Development of Lymphomas

The relationship between inflammatory activity and lymphoma risk is clear [58]. Cytokines such as interleukin-2 (IL-2), IL-5, IL-6, IL-10, and tumor necrosis factor-α (TNF-α) play a key role here. In particular, IL-10 is important because the presence of high levels of this cytokine can alter the delicate balance that exists between T helper cells 1 (Th1) and 2 (Th2). Antigen stimulation and chronic inflammation may expose the lymphoid cell to an increased risk of genetic changes, with consequent clonal expression and development of lymphoma [59]. There is also a greater incidence of autoimmune diseases in the course of lymphoma. The pathogenic mechanisms involved are not clear; however, previous studies in the literature indicate the role of impaired cellular and humoral immunity [3].

There are many theories regarding autoimmunity as a risk factor for developing lymphoma. Initially, the assumptions focused on the similar processes of lymphocyte proliferation, which characterize both autoimmune diseases and hematological malignancies [60]. According to the analysis, both types of diseases are the result of multi-stage processes during which checkpoints that inhibit the uncontrolled growth of B-lymphocytes, including the uncontrolled growth of autoimmune lymphocytes, are eliminated. The most appropriate example is that somatic and germline FAS mutations are associated with both autoimmune diseases and lymphomas. These mutations deregulate apoptosis and increase lymphoid hyperplasia [61,62,63].

Many patients with lymphoma have an autoimmune disease. The most common are Sjögren’s syndrome, thyroiditis, polymyositis, scleroderma, rheumatoid arthritis (RA), vasculitis, autoimmune hepatitis, autoimmune hemolytic anemia (AIHA), and systemic lupus erythematosus (SLE) [7,9].

Swiss et al. found that about one-third of lymphoma patients have autoantibodies typical for autoimmune diseases. In addition, the prevalence of autoantibodies is greater in patients with NHL than in patients with HL. The most common antibodies detected were autoantibodies directed against single-stranded DNA (ssDNA), RPA ribonucleoprotein, Smith antigen (Sm), Sjögren syndrome antigens (SSA and SSB), and cardiolipin [64]. Additionally, Gyumors found that antinuclear antibodies are more frequently detected in NHL patients than in controls [65].

Other theories indicate that a common feature of autoimmune diseases may be disorders in the activity of regulatory T (T_reg_) cells. Interestingly, in NHL, T_reg_ cells attenuated CD8 T-cells, thereby protecting lymphoma cells from cytotoxic effects. However, in some individuals, lymphoma cells are killed by T_reg_ cells. In such cases, the decreased number of these cells may explain both the autoimmunity and the aggravation of the lymphoma [66,67].

A study by Kuksin et al. investigated the role of the NOTCH1 and NOTCH2 pathways, which are deregulated in both diseases. The chronically active NOTCH pathway is involved in the maturation of T-lymphocytes; however, constant inflammation can lead to damage of the immune system and trigger the process of lymphomagenesis [68].

### 4.2. Effect of Viral Infection on Increased Risk of Lymphoma during Immunosuppressive Therapy

The decision to treat patients with a history of lymphoma should be weighed against the potential risk for an Epstein–Barr virus (EBV) infection and the development of LPD in this patient group. Epstein–Barr virus has been associated with some lymphomas [69], which may be particularly important when receiving treatment with MTX [70]. It has been demonstrated that patients can develop MTX-associated LPD. Interestingly, these lymphomas resolve spontaneously after discontinuation of MTX in most cases. Additionally, there is speculation that MTX may reactivate latent EBV.

Treatment with thiopurine may increase the risk for lymphoma development, especially in young, seronegative, EBV-positive men (under 30 years of age). In addition, it is suspected that there is an increased risk for hepatosplenic lymphoma, especially in patients receiving combination therapy [30,69,71,72].

In EBV-negative men, it may be reasonable to avoid thiopurine and consider alternative immunosuppressive therapies [73]. Moreover, in those who develop new onset mononucleosis, thiopurine therapy is interrupted and never restarted.

There are currently no recommendations for viral infection testing; however, it appears to be an appropriate course of action prior to the initiation of immunosuppressive therapy.

There are an increasing number of recommendations from single case studies to test young men for EBV. In patients who have a negative EBV test result, the use of MTX as an immunomodulator in the place of thiopurines is suitable. We have based this procedure on recommendations from transplantology studies [74].

Transplantologists have developed a number of strategies to reduce the risk and consequences of proliferation associated with EBV. For example, measurement of the systemic EBV viral load, which may indicate the need to improve immunosuppression in the case of a significant increase, or the need for preventive treatment such as rituximab, antiviral therapy, or cytotoxic T cells. However, the use of similar treatment regimens in IBD patients remains questionable [22,75].

Problems could also arise with other viruses, which may induce diseases other than lymphoproliferative neoplasms and impair the quality of life in IBD patients. Infection with cytomegalovirus (CMV) can cause the acute exacerbation of the basic disease, while the retention of thiopurines improves the patient’s health [76]. In addition, there is a strong association between CMV infection and NHL; however, CMV may also occur as a complication of lymphoproliferative tumor treatment [71]. Acute chickenpox, also known as varicella zoster virus (VZV), can be fatal in patients receiving thiopurines, and therefore, their use should be discontinued once VZV infection is diagnosed. For herpes zoster infection, there are no recommendations for discontinuing immunosuppressive treatment. Moreover, the reactivation of hepatitis B in patients receiving anti-TNF treatment may lead to liver failure [76]. The exacerbation of human papillomavirus (HPV)-induced lesions may occur when immunosuppression is used, in which case thiopurines are only withheld when the lesions are numerous [76].

Considering the above factors, it is appropriate to consider the effect of individual (selected) drug classes on lymphoma development (Table 2). Consideration should be given to whether the risk of relapse of lymphoproliferative disorders is higher in patients who have previously been treated for lymphoma.

### 4.3. The Role of IBD Therapy in Increasing the Risk of Cancer Recurrence

The management of IBD in patients with a history of lymphoma can be challenging and requires the close coordination of care between gastroenterologists and hematologists/oncologists. Potential clinical issues necessitating special consideration include the use of immunosuppressive therapies for IBD, in particular, biologic agents that inhibit cytokine and immune pathways, which might lead to a relapse of the hematological disease (Table 2). This is of particular interest in patients with active lymphoma as well as in those with a history of hematological malignancy. Therefore, clinical decision-making must include a careful analysis of risks and benefits, with special attention given to prognosis and life expectancy. Below, we attempt to explore these clinical issues, taking into consideration most of the currently available IBD therapies.

There are few studies regarding the treatment of CD in patients with a history of lymphoma. However, it is well-recognized that IBD patients are at an increased risk for developing another cancer, with the risk being six times greater if the first cancer occurred in childhood [77]. The Cancers Et Surrisque Associé aux Maladies inflammatoires intestinales En France (CESAME) study evaluated the risk for new cancer development or recurrence in patients with IBD and a history of cancer, depending on whether or not they were receiving immunomodulator drugs. The authors concluded that in patients with a history of cancer receiving IBD treatment, immunosuppressive therapy has no effect on this risk [77]. In other cases, the study groups were too small to draw conclusions about the risk in patients treated with methotrexate (MTX) or anti-TNF drugs. It has been estimated that the risk of developing a second neoplasm increases 11-fold after 9.5 years of treatment with anti-TNF drugs in patients with a history of melanoma, and therefore, the use of these drugs should be avoided in this patient group [78].

This issue was raised by Mourabet and his colleagues in 2005 in a publication describing nine CD patients with a history of lymphoma. However, in eight of those described cases, lymphoma was diagnosed during the course of CD therapy. Because of this, only one case described a diagnosis of lymphoma preceding the diagnosis of CD by several years—this was the case of a young woman who was diagnosed with CD nine years after oncological therapy. The remaining cases described patients who had previously received either infliximab (IFX) or immunosuppressive treatment or those patients in whom the diagnosis of CD coincided in time with the diagnosis of lymphoma. Nevertheless, their study attempted to determine the optimal therapy for these patients. Various therapies were used, with only a single case reporting the recurrence of lymphoma after 3 years [33]. Moreover, there are many reports in the literature involving CD patients who have been diagnosed with different types of lymphoma. In 60% of cases, lymphomas occur in the small and large intestines [18,79,80,81]. However, it should be noted that information regarding the safety of IBD therapy in those with a history of lymphoma is scarce.

Unfortunately, clinical studies have not yet clarified whether CD therapy is safe when used in patients who have received oncological treatment. By assessing the risk of particular drugs, these patients are eliminated from observational studies.

Until now, there have been a limited number of studies regarding the treatment of IBD in patients with a history of lymphoma. This might be due to the fact that in routine practice, physicians do not introduce intensive pharmacotherapy because of possible adverse effects.

Therefore, treatment is usually limited to the use of mesalazine, nutritional therapy, or periodic steroid therapy [74,82], which can lead to the worsening of IBD and numerous complications requiring surgical treatment. Ultimately, this significantly decreases the quality of life in these patients.

The main dilemma is choosing an effective and safe therapy, with benefits which outweigh the possible risks. It is very important to determine which biological drugs are most effective in these cases. It is currently unclear whether immunosuppression should always be used or when it is appropriate to discontinue intensive treatment.

### 4.4. Mechanism of Thiopurine Leading to Possible Relapse of Lymphoma

Thiopurine derivatives interfere with the synthesis of nucleic acids. In this way, these drugs prevent the proliferation of cells involved in the immune response [27,75]. Thiopurines induce and maintain remission, increase quality-adjusted life expectancy, and facilitate steroid withdrawal [49,83].

These drugs are associated with an increased risk for lymphoma in rheumatoid arthritis (RA) and transplant patients and, therefore, can induce lymphoproliferative tumors in IBD patients [29,84]. As a result of the inhibition of cytotoxic T-lymphocytes and natural killer cells (NK) by thiopurines, immune surveillance is impaired, leading to the development of LPD and is perhaps partly related to the proliferation of EBV-infected lymphocytes [22,26,35,69,85,86]. It may also be associated with the apoptosis of activated T-cells [69]. Thiopurine analogs can also lead to a reduction in the activity of the DNA repair mechanism, resulting in a greater number of mutations. Additionally, there is a lack of regulation of clonal expansion, which leads to LPD and may be the basis of lymphomas [22,74]. Their lymphomagenic effect is reversible, and the function and number of cells return to normal after treatment discontinuation [86]. In addition, the risk for myelotoxicity appears to be approximately five times greater with mercaptopurine than with azathioprine (AZA) [87]. Studies have shown that in IBD patients treated with AZA or mercaptopurine, there is an approximately four times greater risk for developing lymphoma, with an even greater risk being associated with male sex, smoking, and age [27,49].

It appears that the risk for developing lymphoma is increased only during immunosuppression and that this risk returns to a level similar to that of the general population after the discontinuation of treatment [21,27].

A greater incidence of NHL is observed in younger patients (up to 30 years of age). However, this has not been clearly established in CD patients, in contrast to the case of RA patients [26,27,84]. At the same time, this risk increases with age, while in patients over 50 years of age, the risk depends on the duration of therapy [26].

A study using the Markov model showed that the preferred alternative therapy should be recommended if the risk for thiopurine-associated lymphoma was greater than 9.8 times above normal, assuming that the disease itself is not associated with the risk of lymphoma [27]. According to retrospective studies, the risk of developing lymphoma increases to approximately four times above normal when receiving thiopurine [21].

Additionally, they found that thiopurine derivatives increase the risk of developing lymphoma. They argued that the severity of IBD may interfere with the increased risk for lymphoma during immunosuppressive therapy. Patients are more likely to develop lymphoproliferative tumors; however, this can be affected by both disease activity and treatment. Concerns regarding potentially fatal complications mean that both patients and clinicians are reluctant to initiate thiopurine treatment [28].

### 4.5. Effect of Methotrexate on the Development of Lymphoproliferative Disease

MTX works by inhibiting the activity of dihydrofolate reductase, which is involved in the conversion of dihydrofolate to tetrahydrofolate. By acting in the S phase of the cell cycle, MTX blocks DNA synthesis and repair, as well as cell replication. Used in high doses, it has cytotoxic and antiproliferative effects by inhibiting dihydrofolate reductase, thereby blocking DNA and RNA synthesis. At low doses, MTX acts as an immunomodulator without cytotoxic or antiproliferative effects [4,88]. The anti-inflammatory effect is likely to increase the concentration of adenosine, which in turn inhibits the production of the inflammatory factors leukotriene B4 (LTB4), TNF-α, interleukin-6 (IL-6), and IL-8, and also increases the synthesis of IL-10 and IL-receptor antagonists. In addition, adenosine is likely to have an inhibitory effect on neutrophil chemotaxis and neutrophil adhesion to endothelial cells [89]. MTX is cytotoxic, induces leukopenia, and is associated with the impaired immune control of B-cell proliferation [69].

Patients with immunodeficiency may develop MTX-associated LPD [90]. The most frequently reported lymphoproliferative neoplasms in MTX-treated patients are DLBCL (35–60%) and classical HL (12–25%) [91].

Evidence for the carcinogenic effect of MTX is a regression of LPD after its discontinuation. However, studies have shown that the disease itself (using the example of an RA patient) may be a risk factor for LPD (a 2–2.5 times greater risk than in the general population). It has been demonstrated that MTX can accelerate the development of LPD, including via the activation of EBV [70,71,80]. A recent report by Feng et al. has shown that MTX directly induced EBV reactivation after infection by releasing infectious virions [71,92]. Most often, disorders resolve after MTX discontinuation, but in some cases, aggressive chemotherapy is required. Despite initial remission after MTX discontinuation, LPD can recur or persist [91].

Interestingly, although its negative effect on the immune system is clear, it appears to be a safer drug than thiopurine and is the first choice for immunosuppressive treatment in young, EBV-seronegative males [19].

### 4.6. The Impact of Selected Biologics on the Risk of Developing Lymphoma in Patients with IBD

#### 4.6.1. Anti-TNF

Infliximab (IFX) and adalimumab (ADA) inhibit the activity of TNF-α, leading to the reduced infiltration of inflammatory cells, chemotaxis, and tissue degradation [29,93].

IFX is a combination of the variable regions from murine anti-human TNF monoclonal antibodies with human IgG1 κ light chains [69]. Since the antibodies originate from animals, treatment may be associated with a greater risk of allergic disease and loss of response due to the production of anti-IFX antibodies [94]. Treatment with IFX has been associated with a decline in lymphocyte counts, leading to the emergence of opportunistic infections [95]. Biological preparations are used in patients whose disease has not been controlled by thiopurines [95]. Malignant transformation is believed to be multi-stage in patients with IBD, following treatment with IFX to a cytokine that is involved in modulating the immune system and inhibiting carcinogenicity. The use of drugs that antagonize this cytokine may promote the neoplastic process [29]. The anti-TNF-α mechanism of action inhibits apoptosis and thus enables the proliferation of neoplastic cells, including lymphomas (Table 3) [22,74]. The use of anti-TNF drugs in combination with immunomodulators is associated with an increased risk for NHL in adult CD patients, but the incidence of these events remains low and should be weighed against the corresponding treatment benefit [96].

Lemaitre et al. estimated the significant relative risk for lymphoma in all treatment groups compared to non-irradiated patients [49].

Studies have shown that patients with initially elevated gamma-delta (γδ-T) cell levels may be more likely to develop LPD while receiving anti-TNF therapy. In a group of young men with elevated baseline γδ-T values, levels of these cells increased significantly after IFX was infused over a 24 h period. However, this was not observed with ADA treatment. Moreover, there was no increase in lymphocyte levels after switching from IFX to ADA [97]. On the other hand, the Food and Drug Administration (FDA) reports an approximately three-fold increase in the risk for lymphoma in patients treated with ADA when compared to the general population [69].

However, subsequent studies have shown that the relationship between anti-TNF drugs and increased risk for lymphoma is still questionable and difficult to demonstrate, including the fact that most patients were exposed to other medications, mainly thiopurines [29]. This suggests that observational studies that have shown an association between lymphoma and anti-TNF have methodological flaws and that the results remain biased. This means that if the diagnosis of lymphoma was made a few days after initiating anti-TNF drug treatment, the adverse event was attributed to the anti-TNF group, even though the result was not reliable. Indeed, the pathophysiological mechanisms supporting cancer development are extremely variable between different types of cancer, so it is unlikely that one drug can trigger all of these mechanisms [29].

Anti-TNF drugs are superior to AZA in previously untreated CD patients and are approved for patients with moderate to severe CD or UC when the disease is not controlled by thiopurines [27,49].

Combination therapy with thiopurines and anti-TNF agents seems to be more effective than monotherapy in IBD patients who have not been previously treated with thiopurines or anti-TNF drugs [49]. In adults, the use of thiopurine monotherapy or anti-TNF therapy alone has been associated with an increased risk of developing lymphoma when compared to patients who are not receiving these drugs. Furthermore, this risk was greater with combination therapy [49]. Data from several studies showed that the risk of developing lymphoma was not statistically significant in patients receiving ADA or IFX [49,95]. Evidence suggests that combination therapy increases the risk for serious opportunistic infections. Additionally, an increased incidence of hepatosplenic T-cell lymphoma was observed [98,99]. Other studies suggest that combination therapy only slightly increases the risk for lymphoma when compared to thiopurine monotherapy [22,76,96]. However, it is recommended to avoid thiopurines in patients with a history of NHL, in which case MTX is preferable [76].

#### 4.6.2. Ustekinumab

Ustekinumab is a fully human IgG1 monoclonal antibody against the p 40 µL12/µL23 subunit. Its action leads to the weakened binding of these interleukins to receptors and to 12Rbeta1 on the surface of T- and NK-lymphocytes, thus weakening the activation of immune cells. In experimental animal models, it has been shown that blocking IL-12 and IL-23 may result in an increased incidence of tumors (Table 3). However, this relationship requires further observation and research [100]. Several investigations found no evidence of an increased incidence of lymphomas when compared to the general population [101,102]. More extensive studies investigated psoriasis patients and found that the frequency of lymphoma in this group was comparable to the general population. However, there are single case reports of lymphoma development following treatment with ustekinumab [98].

### 4.7. The Role of Anti-Integrin (Vedolizumab) on Lymphoma Development

Vedolizumab is a humanized immunoglobulin G1 monoclonal antibody to a4B7 integrin [103]. Its role is to selectively prevent the migration of leucocytes into the gastrointestinal submucosa [104], which inhibits the interaction of T cells, monocytes, and dendritic cells with the mucosal location in the cell adhesion molecule-1 (Mad-CAM-1) expressed on vascular endothelium. The rate of malignancy during the therapy is consistent with that normally observed in patients with IBD [104]. Clinical studies suggest that vedolizumab does not have a systematic immunosuppressive effect and does not increase the risk of malignancy. However, there are reports that by inhibiting the interaction between the a4b7 MAdCAM-1 integrin, the number and function of circulating T lymphocytes may be reduced, also through the interactions of chemokines and cytokines, for example, IL-23A and IL-17 [105].

### 4.8. The Role of Small Molecules (JAKs) on Lymphoma Development

Janus kinase inhibitors (JAKs) are orally administered small molecules designed to block cytokines, preventing phosphorylation of Janus kinases with the cytokine receptor. Janus kinase inhibitors target key cytokines in the pathogenesis of IBD. Janus kinase inhibitors, such as tofacitinib, act simultaneously on many cytokines, leading to a broader inhibition of chronic inflammatory processes, including the inhibition of the JAK3-STAT5 pathway [36,94,95]. Mutations of JAK3 have been shown to occur in hematological malignancies, and studies have shown that tofacitinib may play a positive role in treatment, especially in EBV(+) patients. It has been found that the inhibition of the JAK3/STAT5 pathway induces the apoptosis of tumor cells. In a study by Ando et al., tofacitinib was shown to induce G1 cell cycle arrest and inhibit tumor growth in EBV-associated T lymphoma and NK-cell lymphoma [36]. Additionally, these drugs are administered orally, which is very convenient for the patient. Moreover, because these agents are not antibodies, they are non-antigenic, meaning that there is no need to use immunomodulators concurrently to prevent the production of antibodies [106].

Taken together, JAK inhibitors appear to be a safer form of treatment as they interfere less with the human immune system [107,108,109,110,111,112]. In terms of adverse lymphoproliferative effects, one study showed the estimated risk for lymphoma to be 0.07 per 100 patient years [113].

New immunosuppressive drugs could be important treatment options for patients with a history of lymphoma. These agents are potentially safer, owing partly to their increased selectivity. There are also reports in the literature regarding the use of allogeneic hematopoietic stem cell transplants (HSCT) or intraperitoneal administration of autologous tolerogenic dendritic cells. These therapies have been used in refractory or comorbid cases, and the results have been very promising [77,94,114].

## 5. Conclusions

The treatment of CD patients who have previously received chemotherapy is a major challenge for gastroenterologists. Both immunosuppression and biological treatments are controversial. There are no unequivocal reports on the safety of therapy in these patients in the literature. Treatment is difficult and requires individual assessment and cooperation with a specialist in hematology–oncology and gastroenterology. However, chronic inflammation can increase the risk of developing DLBCL localized in the gastrointestinal tract. Further studies are needed to formulate appropriate recommendations.

## Figures and Tables

**Figure 1 ijms-25-04241-f001:**
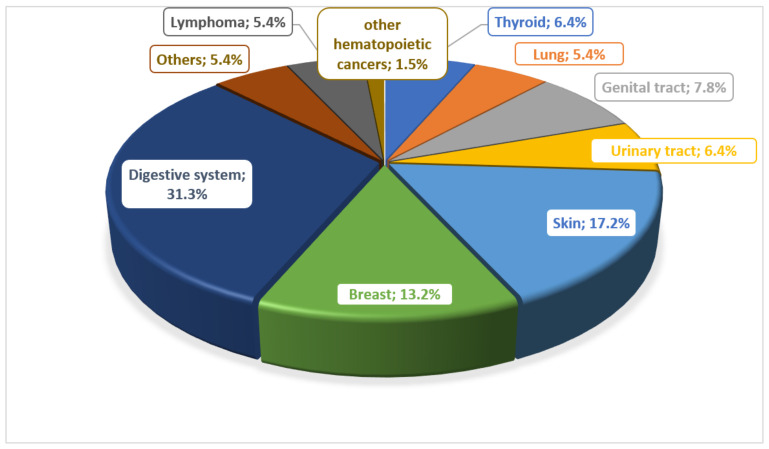
Prevalence of different types of cancer in patients with Crohn’s disease.

**Figure 2 ijms-25-04241-f002:**
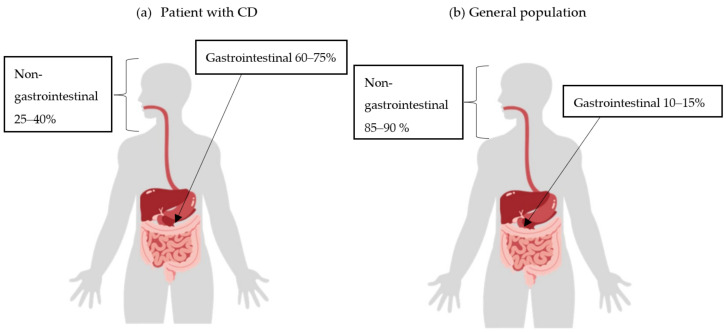
Incidence of extranodal non-Hodgkin lymphoma (NHL) by location in Crohn’s disease (CD) patients (**a**) and the general population (**b**).

**Table 1 ijms-25-04241-t001:** Incidence of lymphoma based on a literature review, 2010–2022.

References	Date of Publication	Total Number of Patients	Diagnosis	Number of Lymphoma Cases	Partition
Abbas et al. [37]	2012	36,826	UC	189	184 cases of NHL 5 cases of HL
Biancone et al. [38]	2016	44,619	CD: 21,953 UC: 22,666	6	NHL: 4 HL: 2 Only in patients with CD
Biancone et al. [17]	2020	403	CD: 204 UC: 199	11	Lymphoma only in patients with CD
Campos et al. [39]	2013	1607	CD: 804 UC: 803	3	CD: 2 UC: 1
Chiorean et al. [40]	2011	3585	IBD	8	CD: 5 UC: 3
de Francisco et al. [41]	2018	1483	IBD	6	CD: 5 UC: 1
Herrinton et al. [25]	2011	16,023	IBD	33	
Jess et al. [42]	2013	2211	CD: 774 UC: 1437	15	CD: 7 UC: 8
Jung et al. [43]	2017	15,921	CD: 5506 UC: 9785	5	CD: 3 UC: 2
Jussila et al. [44]	2013	20,970	CD: 4983 UC: 15,987	72	CD: 14 UC: 58
Khan et al. [23]	2013	36,891	UC	119	
Khan et al. [45]	2017	63,759	CD: 28,322 UC: 35,437	65	
Kobayashi et al. [46]	2020	75,673	IBD	103	
Kopylov et al. [47]	2015	19,582	IBD	121	
Lakatos et al. [48]	2012	1420	CD: 506 UC: 914	3	CD: 1 UC: 2
Lemaitre et al. [49]	2017	189,289	CD: 95,537 UC: 93,752	336	
Madanchi et al. [50]	2016	22	CD: 13 UC: 9	5	CD: 4 UC: 1
Mizushima et al. [51]	2010	294	CD	0	
Pasternak et al. [52]	2013	38,772	IBD	46	
Ranjan et al. [53]	2022	1093	CD: 305 UC: 788	0	
Scharl et al. [54]	2019	3119	CD: 1777 UC: 1342	103	
van den Heuvel et al. [55]	2016	2801	IBD	3	
van Domselaar et al. [56]	2010	911	IBD	7	
Yano et al. [57]	2013	770		0	

**Table 2 ijms-25-04241-t002:** The frequency of adverse effects in terms of hematologic disorders and the lymphatic system in particular groups of drugs.

Group of Drugs	Frequency of Adverse Effects in Terms of Hematologic and Lymphatic System Disorders
Azathioprine	Leukopenia—≥1/10 patients Thrombocytopenia—≥1/100 to <1/10 patients Granulocytopenia, anemia—≥1/10,000 to <1/100 patients Agranulocytosis, pancytopenia, aplastic anemia—<1/10,000 patients Lymphoproliferative disorders—≥1/10,000 to <1/1000 patients
6-Mercaptopurine	Leukopenia, thrombocytopenia—≥1/100 to <1/10 patients Megaloblastic anemia—≥1/10,000 to <1/1000 patients Acute myeloblastic leukemia—<1/10,000 patients Lymphoproliferative disorders—≥1/10,000 to <1/1000 patients
Methotrexate	Leukopenia, thrombocytopenia, anemia—≥ 1/100 to <1/10 patients Pancytopenia, agranulocytosis—≥1/1000 to <1/100 patients Malignant lymphoma—≥1/1000 to <1/100 patients
Infliximab	Neutropenia, leukopenia, anemia—≥1/100 to <1/10 patients Thrombocytopenia, lymphopenia, lymphocytosis—≥1/1000 to <1/100 patients Agranulocytosis, pancytopenia, hemolytic anemia—≥1/10,000 to <1/1000 patients Lymphoma, Non-Hodgkin lymphoma, Hodgkin lymphoma, leukemia—≥1/10,000 to <1/1000 patients Hepatosplenic T-cell lymphoma—frequency unknown
Adalimumab	Lymphoma—≥1/1000 to <1/100 patients Leukemia—≥1/10,000 to <1/100 patients Hepatosplenic T-cell lymphoma—frequency unknown Leukopenia, anemia—≥1/10 patients Leukocytosis—≥1/100 to <1/10 patients Pancytopenia—≥1/10,000 to <1/100 patients
Tofacitinib	Leukopenia, anemia—≥1/100 to <1/10 patients Lymphopenia, neutropenia—≥1/10,000 to <1/100 patients

**Table 3 ijms-25-04241-t003:** Selected action mechanisms of drugs that may cause lymphoproliferative disorders.

Drugs	Thiopurines	Methotrexate	Anti-TNF	Ustekinumab
Action mechanisms of drugs that may be common to lymphoproliferative disorders	✓ Inhibition of cytotoxic lymphocyte (Tc) and NK cells ✓ proliferation of EBV-infected cells ✓ apoptosis of activated T lymphocytes ✓ inhibition of DNA repair mechanisms	✓ inhibition of DNA synthesis and repair ✓ inhibition of LTB4, TNF-α, IL-6, and IL-8 ✓ increases the synthesis of IL-10 ✓ impaired immune control of B cells ✓ EBV activation	✓ inhibition of cellular apoptosis	✓ weakening of T and NK cell activity ✓ inhibition of IL-12 and IL-23

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
