# Peer review of "The Complex Relationship between Mechanisms Underlying Inflammatory Bowel Disease, Its Treatment, and the Risk of Lymphomas: A Comprehensive Review"

_ijms, 2024, doi:10.3390/ijms25084241_

Round 1
Reviewer 1 Report (New Reviewer)
Comments and Suggestions for Authors
This review article aims to describe the causative relationship between different immunosuppressive agents used in the treatment of Inflammatory Bowel Disease and the risk of developing lymphoproliferative disorders. The topic is very interesting and relevant information is lacking from the literature. At a first sight, the authors appear to provide a well-structured manuscript using an extensive reference list. However, when the sub-chapters are considered, the structure and the flow of the text is clearly suboptimal. Furthermore, the use of English language is problematic. Overall, the text is very difficult to follow and needs substantial revision in the presentation of the scientific data as well as major language editing.
Comments on the Quality of English LanguageAs stated above, the text needs major language editing.
Author Response
Thank you very much for your review. As suggested, we corrected the linguistic errors that appeared in the text. In addition, we changed the structure a bit. In the first part, we described the characteristics of Crohn's disease, UC and Lymphomas. Then we added the mechanisms causing cancer development in IBD, then lymphoma development in IBD. In the second part of the manuscript, on the other hand, we wanted to consider how the various known risk factors might influence the development of lymphoma in IBD patients. We hope the structure is clearer.
Reviewer 2 Report (New Reviewer)
Comments and Suggestions for Authors
The authors present a manuscript investigating relevant clinical issues related to the treatment of patients with inflammatory bowel disease and active lymphoma, as well as patients with a hematologic history of cancer.
The findings indicate that treating patients with Crohn's disease (CD) who have previously undergone chemotherapy poses a significant challenge for gastroenterologists. Both immunosuppression and biological treatments have been subject to controversy, with inconclusive evidence regarding their safety in these circumstances. Managing the condition is complex and requires an individualized approach, as well as close collaboration between specialists in hematology-oncology and gastroenterology. It is crucial to note that chronic inflammation may increase the risk of developing diffuse large B-cell lymphoma (DLBCL) localized in the gastrointestinal tract. Further studies are needed to establish appropriate therapeutic guidelines.
However, it is important to note that in Figure 2, the acronym NHL (Non-Hodgkin Lymphoma) is not explained. It is essential that figure and table captions are self-explanatory. Additionally, Figure 1 could be supplemented with a more detailed caption that better describes the content presented.
Author Response
Thank you very much for appreciating the topic we described. As recommended, the acronym NHL has been clarified. In addition, we have also changed the caption of Figure 1, it seems that now it will be much better understood.
Reviewer 3 Report (New Reviewer)
Comments and Suggestions for Authors
ijms-2910726
Type of manuscript: Review
Title: Complex relationship between mechanisms underlaying inflammatory bowel disease and it treatment and the risk of lymphomas. A comprehensive review.
Authors: Katarzyna Stasik, Rafał Filip*
This paper is a review article on the inflammatory bowel disease and lymphomas. Overall, it is well-written; however, it requires more detailed corrections and updates.
[Major concerns]
1. Title: The current title of the paper is not grammatically appropriate in English, so please review it again.
2. It's fortunate that the authors' affiliated research lab primarily focuses on IBD. As you are well aware, CD and UC are the predominant conditions within IBD, and it's known that various cancers can arise from these two diseases. However, the authors predominantly mention CD, providing insufficient coverage on UC. In my opinion, it's essential to address this aspect first, particularly emphasizing that lymphoma occurs more frequently in CD patients compared to those with UC.
3. Moreover, it is well known that the molecular mechanisms underlying the occurrence of colorectal cancer associated with IBD differ significantly from sporadic colorectal cancer. Introducing such information, it is necessary to describe more specifically the mechanisms underlying the occurrence of lymphoma associated with IBD.
4. Abbreviations: The use of abbreviations when writing a paper has many advantages besides simplicity of expression. To use an abbreviation, first write the abbreviation in parentheses after the full name, and then use the abbreviation from Introduction to the final Conclusion. Only in Abstract and Figure legend do it separately. If an abbreviation is not used more than twice, there is no need to define it, so please delete it.
5. English: The English composition of the paper is generally well-done. However, some names of disease or compound names are written in uppercase letters even though they are not the first letter of the sentence or proper nouns. Please make corrections throughout the text and in the figures. Additionally, the notation of certain words is inconsistent. Examples: IL-6 vs. IL 6 vs. Il 6, etc.
6. Table 3 is missing. The second Table 2 might be Table 3.
[Minor concerns]
1. Line 11: Define anti-TNF in the abstract.
2. Line 41: ‘Non-Hodgkin’ should be written as ‘non-Hodgkin’.
3. Line 57: Crohn’s Disease should be written as Crohn’s disease.
4. Line 71: Re-write the full name of ‘leuko-’.
5. Line 73: Define CLL.
6. Line 92: Define EBV here.
7. Line 97: Re-write this line including [Fig. 1].
8. Table 1: ‘Biancone et al’ should be written as ‘Biancone et al.’.
9. Table 2: ‘6-mercaptopurine’ should be written as ‘6-Mercaptopurine’ as other compounds. When an Arabic numeral follows the initial letter of a compound, the subsequent letter should be capitalized.
10. Table 2: In Table 2, in the right column describing Methotrexate, Infliximab, and Adalimumab, there are four instances where "do" is used instead of "to," which seems to be a mistake. Please verify this and correct it if necessary.
11. Line 169: Define CESAME study.
12. Line 210: Define LPD.
13. Line 246: Define HPV.
14. Line 260: Define NK.
15. Line 292: Methotrexate should be written as MTX.
16. Line 340: Define and explain y6T.
17. Line 345: Define FDA.
18. Lines 378 and 379: Correct IL 12 and IL 23.
19. Line 388: Table 3, not Table 2.
20. Table 3: Metotrexate should be written as Methotrexate.
21. Table 3: Define Tc. TNFα should be written as TNF-α. Re-write Il6, Il8, Il 10, Il 12, etc.
22. Line 399: Re-write IL23A.
23. Line 400: Re-write IL17.
24. Line 410: ‘Shotero Ando et al.’ Is ‘Shotero Ando’ the last name the first author? If not, just write the last name only.
25. Numbering of references in the main text: The citation style for references within the main body of the paper does not adhere to the IJMS style. Please correct this.
Overall, the manuscript can be considered to publication after major revision as indicated above.
Comments on the Quality of English Languageijms-2910726
Type of manuscript: Review
Title: Complex relationship between mechanisms underlaying inflammatory bowel disease and it treatment and the risk of lymphomas. A comprehensive review.
Authors: Katarzyna Stasik, Rafał Filip*
This paper is a review article on the inflammatory bowel disease and lymphomas. Overall, it is well-written; however, it requires more detailed corrections and updates.
[Major concerns]
1. Title: The current title of the paper is not grammatically appropriate in English, so please review it again.
2. It's fortunate that the authors' affiliated research lab primarily focuses on IBD. As you are well aware, CD and UC are the predominant conditions within IBD, and it's known that various cancers can arise from these two diseases. However, the authors predominantly mention CD, providing insufficient coverage on UC. In my opinion, it's essential to address this aspect first, particularly emphasizing that lymphoma occurs more frequently in CD patients compared to those with UC.
3. Moreover, it is well known that the molecular mechanisms underlying the occurrence of colorectal cancer associated with IBD differ significantly from sporadic colorectal cancer. Introducing such information, it is necessary to describe more specifically the mechanisms underlying the occurrence of lymphoma associated with IBD.
4. Abbreviations: The use of abbreviations when writing a paper has many advantages besides simplicity of expression. To use an abbreviation, first write the abbreviation in parentheses after the full name, and then use the abbreviation from Introduction to the final Conclusion. Only in Abstract and Figure legend do it separately. If an abbreviation is not used more than twice, there is no need to define it, so please delete it.
5. English: The English composition of the paper is generally well-done. However, some names of disease or compound names are written in uppercase letters even though they are not the first letter of the sentence or proper nouns. Please make corrections throughout the text and in the figures. Additionally, the notation of certain words is inconsistent. Examples: IL-6 vs. IL 6 vs. Il 6, etc.
6. Table 3 is missing. The second Table 2 might be Table 3.
[Minor concerns]
1. Line 11: Define anti-TNF in the abstract.
2. Line 41: ‘Non-Hodgkin’ should be written as ‘non-Hodgkin’.
3. Line 57: Crohn’s Disease should be written as Crohn’s disease.
4. Line 71: Re-write the full name of ‘leuko-’.
5. Line 73: Define CLL.
6. Line 92: Define EBV here.
7. Line 97: Re-write this line including [Fig. 1].
8. Table 1: ‘Biancone et al’ should be written as ‘Biancone et al.’.
9. Table 2: ‘6-mercaptopurine’ should be written as ‘6-Mercaptopurine’ as other compounds. When an Arabic numeral follows the initial letter of a compound, the subsequent letter should be capitalized.
10. Table 2: In Table 2, in the right column describing Methotrexate, Infliximab, and Adalimumab, there are four instances where "do" is used instead of "to," which seems to be a mistake. Please verify this and correct it if necessary.
11. Line 169: Define CESAME study.
12. Line 210: Define LPD.
13. Line 246: Define HPV.
14. Line 260: Define NK.
15. Line 292: Methotrexate should be written as MTX.
16. Line 340: Define and explain y6T.
17. Line 345: Define FDA.
18. Lines 378 and 379: Correct IL 12 and IL 23.
19. Line 388: Table 3, not Table 2.
20. Table 3: Metotrexate should be written as Methotrexate.
21. Table 3: Define Tc. TNFα should be written as TNF-α. Re-write Il6, Il8, Il 10, Il 12, etc.
22. Line 399: Re-write IL23A.
23. Line 400: Re-write IL17.
24. Line 410: ‘Shotero Ando et al.’ Is ‘Shotero Ando’ the last name the first author? If not, just write the last name only.
25. Numbering of references in the main text: The citation style for references within the main body of the paper does not adhere to the IJMS style. Please correct this.
Overall, the manuscript can be considered to publication after major revision as indicated above.
Author Response
Thank you for your detailed review. Thank you very much for the pertinent tips.
1. The title- has been grammatically corrected.
2. We described additional UC briefly, as suggested, we pointed out that lymphoma occurs more often in patients with CD.
3. We have clarified the mechanisms underlying the development of cancer in IBD, the mechanisms of lymphoma development in IBD are also described in the manuscript.
4. Abbreviations have been expanded accordingly.
5. Language errors have been corrected.
6. Captions have been changed.
Minor.
1-24 were made all corrections as recommended.
25. Numbering and citation style have been corrected.
Thank you very much once again for your feedback,
Round 2
Reviewer 3 Report (New Reviewer)
Comments and Suggestions for Authors
[Please see the attached PDF file.]

Comments on the Quality of English Language[Please see the attached PDF file.]
Author Response
Thank you very much for your review. Following your suggestions, we have improved the shortcuts once again. We have reviewed the entire manuscript again and suggested corrections have been made. We hope that the article is now correct and coherent. Thank you very much for your work.
This manuscript is a resubmission of an earlier submission. The following is a list of the peer review reports and author responses from that submission.
Round 1
Reviewer 1 Report
Comments and Suggestions for Authors
This literature review summarizes the current situation of the probable consequences of utilizing the current therapeutic approaches to people who may be diagnosed with Crohn's disease after diagnosis of LPDs, and also shuffles with the probability of Crohn's disease patients getting LPDs.
Although this looks well written and has a good mix of information and a discussion of the current situation, it is confusing in terms of how it is discussed.
1. The title informs us that the review deals with discussing therapeutic approaches in patients who suffer from Crohn's disease who previously had lymphoma. However, immediately, the abstract background seems like IBD will be discussed as a risk factor for lymphomas, however, the methods and conclusion parts stick to the title's message.
2. This is exactly the situation with the body of the manuscript. Introduction deals with how autoimmune diseases and IBD might be considered risk factors for HL , NHL, etc. The introduction concludes by saying most studies have not confirmed this risk, but nevertheless, guidelines for treating severe CD in patients with LPDs have not been established. I do not understand the link here.
Either the review must be titled in a way that reflects both: IBD as a risk factor for GI lymphomas and vice-versa, which would argue their case, or let go of the portions that discuss IBD as a risk factor for LPDs from the whole review.
3. Again, in the discussion (page 3) it starts well by trying to bring common ground between IBD and Lymphomas. The content is relevant knowledge to know about what autoimmune diseases do lymphoma patients have the chance of being diagnosed, and what might be a reason for that.
But then, in the discussion beginning with immunomodulators, it is again about how thiopurines, MTX and biologics that are used to treat IBD- increase the risk factor for lymphomas and how one should consider the risk to be incorporated into IBD treatment plans. But it seems like the exact opposite should have been reviewed, according to the title and the first part of the discussion section.
Recommendations:
In my opinion, changing the title to reflect how and by what mechanisms can IBD as well as IBD treatment affect the risk of being diagnosed with lymphomas and its treatment would be really appropriate.
This should also be accompanied by changing the chronology and course of the discussion.
Certain figures or flowcharts that can depict the immune signaling mechanisms common to IBD and lymphomas can be much more illustrative and make the review easier to understand.
The discussion about the 'vice-versa' situation can be handled separately. The review need not be divided into "Discussion" , it can just discuss all the topics distinctly.
Author Response
Thank you very much for your opinion, pertinent observations, and professional review. As suggested, we have modified the abstract so that it is in line with the assumptions of the review.
We have divided the discussion with subtitles to make it more readable.
We have added a diagram showing the frequency of extranodal NHL in CD patients compared to the general population to better illustrate the problem. In addition, we have included a table summarizing the mechanisms of action of immunomodulators and selected biologic drugs, which may be the basis of lymphoproliferative disorders.
In the review, we wanted to present the impact of different processes, both the mechanism of chronic inflammation and the effects of drugs used in patients with IBD. This is a difficult subject due to the lack of clear recommendations. It, therefore, seems reasonable to present the different processes. Through this review, we would like to facilitate future therapeutic decisions. It seems appropriate to hypothesize that the risk of developing lymphoproliferative disorders will be higher in patients who have previously been treated for lymphoma. If it is possible we would like to change the title too.
Once again, thank you very much for your feedback.
Reviewer 2 Report
Comments and Suggestions for Authors
Here are some suggestions for the authors:
Provide a clearer structure for the discussion of lymphoma in relation to both Ulcerative Colitis (UC) and Crohn's Disease(CD). The aims of the review is not clear. There are some suggestions.
UC, CD or IBD would be reviewed respectively.
Considering on the specific types of lymphoma.
The manuscript touches upon the association between treatment agents and lymphoma. It is recommended to present this association in a more organized manner, perhaps through the use of tables or figures for enhanced clarity.
Comments on the Quality of English LanguageHere are some suggestions for the authors:
Provide a clearer structure for the discussion of lymphoma in relation to both Ulcerative Colitis (UC) and Crohn's Disease(CD). The aims of the review is not clear. There are some suggestions.
UC, CD or IBD would be reviewed respectively.
Considering on the specific types of lymphoma.
The manuscript touches upon the association between treatment agents and lymphoma. It is recommended to present this association in a more organized manner, perhaps through the use of tables or figures for enhanced clarity.
Author Response
Thank you very much for your review. In this review, we wanted to show how different mechanisms (the effect of drugs, chronic inflammation, the effect of viral infections) may affect the development of lymphoma in patients with CD. We also wanted to consider what therapeutic decisions should be made in the case of patients with CD who had previously been treated for lymphoma. We hope that our review will facilitate the therapeutic decision of gastroenterologists in the future. For this reason, the layout of our article seems reasonable.
As suggested, we have added a table summarizing some mechanisms of action of immunosuppressive drugs and biologics underlying the development of lymphoproliferative disorders. If it is possible we would like to change the title. In addition, we have modified the abstract and divided the discussions into subtitles to make it easier to read.
We have added a diagram showing the frequency of extranodal NHL in CD patients compared to the general population to better illustrate the problem.
Once again, thank you very much for your opinion.
Round 2
Reviewer 1 Report
Comments and Suggestions for Authors
Thank you for modifying the title and abstract. I would like to point out that the conclusion in the abstract might be rephrased, and it seems like the sentences are incomplete (line 24 "patients with a history of....").
Comments on the Quality of English Language
English is fine. Just some sentences need to be completed.
Reviewer 2 Report
Comments and Suggestions for Authors
The authors’ find a rare case of lymphoma in a patients with C.D . The relationship between inflammatory bowel diseases (IBD), including Crohn's disease, and lymphoma has not been well studied in the previous literature. Lymphoma in IBD seemed not be with enough evidence even in this review by ref. 8, 24, 43 as the author mentioned.
On the view of chronic inflammation, and immunotherapy for IBD, patients with IBD indeed with risk of cancer, including lymphoma. I suggest the authors re-write with more clear structure in the text of manuscript, including the following parts, cancer in IBD, UC, CD, lymphoma in IBD, UC, CD, and the relation of immune therapy in IBD, UC, and CD. The structure of the abstract should be improved.
Comments on the Quality of English Languagemoderate edit